# CMR Tissue Characterization in Patients with HFmrEF

**DOI:** 10.3390/jcm8111877

**Published:** 2019-11-05

**Authors:** Patrick Doeblin, Djawid Hashemi, Radu Tanacli, Tomas Lapinskas, Rolf Gebker, Christian Stehning, Laura Astrid Motzkus, Moritz Blum, Elvis Tahirovic, Aleksandar Dordevic, Robin Kraft, Seyedeh Mahsa Zamani, Burkert Pieske, Frank Edelmann, Hans-Dirk Düngen, Sebastian Kelle

**Affiliations:** 1Department of Internal Medicine/Cardiology, German Heart Center Berlin, 13353 Berlin, Germany; tanacli@dhzb.de (R.T.); Tomas.Lapinskas@lsmuni.lt (T.L.); gebker@dhzb.de (R.G.); christian.stehning@philips.com (C.S.); zamani@dhzb.de (S.M.Z.); pieske@dhzb.de (B.P.); 2DZHK (German Center for Cardiovascular Research), Partner Site Berlin, 10115 Berlin, Germany; djawid.hashemi@charite.de (D.H.); frank.edelmann@charite.de (F.E.); hans-dirk.duengen@charite.de (H.-D.D.); 3Department of Internal Medicine/Cardiology, Charité Campus Virchow Klinikum, 13353 Berlin, Germany; laura-astrid.motzkus@charite.de (L.A.M.); moritz.blum@charite.de (M.B.); elvis.tahirovic@charite.de (E.T.); aleksandar.dordevic@charite.de (A.D.); robin.kraft@charite.de (R.K.); 4Department of Cardiology, Medical Academy, Lithuanian University of Health Sciences, 50161 Kaunas, Lithuania; 5Philips Healthcare, 22335 Hamburg, Germany

**Keywords:** HFmrEF, T2 mapping, T1 mapping, ECV, fibrosis, inflammation, strain

## Abstract

The characteristics and optimal management of heart failure with a moderately reduced ejection fraction (HFmrEF, LV-EF 40–50%) are still unclear. Advanced cardiac MRI offers information about function, fibrosis and inflammation of the myocardium, and might help to characterize HFmrEF in terms of adverse cardiac remodeling. We, therefore, examined 17 patients with HFpEF, 18 with HFmrEF, 17 with HFrEF and 17 healthy, age-matched controls with cardiac MRI (Phillips 1.5 T). T1 and T2 relaxation time mapping was performed and the extracellular volume (ECV) was calculated. Global circumferential (GCS) and longitudinal strain (GLS) were derived from cine images. GLS (−15.7 ± 2.1) and GCS (−19.9 ± 4.1) were moderately reduced in HFmrEF, resembling systolic dysfunction. Native T1 relaxation times were elevated in HFmrEF (1027 ± 40 ms) and HFrEF (1033 ± 54 ms) compared to healthy controls (972 ± 31 ms) and HFpEF (985 ± 32 ms). T2 relaxation times were elevated in HFmrEF (55.4 ± 3.4 ms) and HFrEF (56.0 ± 6.0 ms) compared to healthy controls (50.6 ± 2.1 ms). Differences in ECV did not reach statistical significance. HFmrEF differs from healthy controls and shares similarities with HFrEF in cardiac MRI parameters of fibrosis and inflammation.

## 1. Introduction

Heart failure is a clinical entity with a diverse spectrum of etiologies and phenotypes. Classification systems and diagnostic criteria have been established paralleling better understanding of its pathophysiology. Early on, the significance of the left ventricular ejection fraction (LV-EF) in the classification of heart failure was acknowledged. The European Society of Cardiology recently suggested to define heart failure with moderately reduced ejection fraction (HFmrEF, LV-EF 40–50%) as a distinct category between heart failure with preserved (HFpEF, LV-EF > 50%) and reduced (HFrEF, LV-EF < 40%) ejection fractions [1]. Patients with HFrEF exhibit functional, structural, cellular and interstitial changes that are summarized as left ventricular remodeling and treatments targeted against remodeling are a mainstay of HFrEF therapy [2]. Unfortunately, these treatments have shown markedly less benefits in patients with HFpEF [3]. Patients with HFmrEF have only recently been proposed as a distinct category, and while debate is still ongoing about whether it truly represents a distinct category or merely a transition zone between HFpEF and HFrEF, available data suggests a possible benefit from treatment against remodeling [4].

Cardiac magnetic resonance (CMR) imaging is a noninvasive method to assess cardiac function, structure, inflammation, and fibrosis. The development of T1 and T2 relaxation time mapping techniques has greatly improved the ability to detect changes in tissue composition, most notably fibrosis and edema [5]. The combination of pre-contrast (native) and post-contrast T1 relaxation time mapping allows an estimation of the extracellular volume (ECV) [6]. While elevations in ECV and native T1 relaxation time seem more related to fibrosis and are strong predictors of adverse outcome, the T2 relaxation time seems more sensitive for the diagnosis of edema and inflammation [7,8,9]. In addition, strain-analysis is a promising new method of functional analysis whose clinical significance is currently still under investigation and which might provide additional prognostic and diagnostic information in heart failure patients [10].

The purpose of our study was to further characterize HFmrEF patients in terms of advanced CMR imaging markers of adverse cardiac remodeling, with a focus on T2 mapping as a potential biomarker in heart failure.

## 2. Experimental Section

From a contemporary trial, a total of 52 well characterized patients with HFpEF, HFmrEF and HFrEF, along with 17 controls, were included in this analysis (EMPATHY-HF, German Clinical Trials Register ID: DRKS00015615) [11]. HFrEF, HFmrEF and HFpEF were defined according to the 2016 ESC guidelines [1]. The study complies with the declaration of Helsinki and was approved by the ethics committee of the Charité-Universitätsmedizin Berlin. All analyses and procedures are covered by the informed consent obtained prior to inclusion. All patients were >45 years, had signs and symptoms of heart failure NYHA II or III (at least 30 d prior to screening), had been stable for at least 7 d (defined as no i.v. diuretics or inotropics, no hospitalization and no medication change). The complete inclusion and exclusion criteria are accessible through the German Clinical Trials Register [11]. All patients received a screening-echocardiography, a cardiac MRI and a comprehensive laboratory evaluation as part of the main study protocol. Quality of life was assessed using the Minnesota Living with Heart Failure Questionnaire. For our analysis, the patients were reclassified based on the results of the cardiac MRI derived LV-EF, leading to 17 patients with HFpEF, 18 with HFmrEF and 17 with HFrEF. When comparing MRI-derived LV-EF to echocardiography-derived LV-EF, roughly one third of HFpEF patients were reclassified as HFmrEF and half of the HFmrEF patients were reclassified as HFrEF.

All patients were examined with a clinical 1.5 Tesla MRI scanner (Achieva, Philips Healthcare, Best, The Netherlands) equipped with a cardiac, five-element phased array coil. Cine images were acquired using a retrospectively gated cine-CMR in cardiac short-axis, vertical long-axis and horizontal long-axis orientations using a steady-state free precession (SSFP) sequence. Native and 15 min post contrast T1-mapping were performed using a modified Look-Locker (MOLLI) 5s(3s)3s-scheme [12]. Typical imaging parameters were as follows: Acquired voxel size = 2.0 × 2.0 × 10 mm^3^, reconstructed voxel size = 0.5 × 0.5 × 10 mm^3^, balanced SSFP readout, flip angle = 35°, parallel imaging (SENSE) factor = 2 and effective inversion times between 150 and 3382 ms. T2-mapping was performed before administration of contrast media using a black-blood-prepared, navigator-gated, free-breathing hybrid gradient (echo planar imaging, EPI) and a spin-echo multi-echo sequence (GraSE), as described previously, with the following typical imaging parameters: TR = 1 heartbeat, 9 echoes (TE_1_ = 15 ms, delta TE = 7.7 ms), FA 90°, parallel imaging (SENSE = 2), EPI factor = 7, black-blood prepulse and breath-hold (scan duration about 14 s). [13] Patients received 0.15 mmol/kg of gadolinium-based contrast agent (Gadobutrol 1.0 mmol/mL, Gadovist^®^, Bayer AG, Leverkusen, Germany). Quantitative modified DIXON-imaging (mDIXON) for late enhancement was performed using a black-blood prepared, T1-weighted, spoiled-gradient, multi-echo sequence with 6 echoes starting 10 min post contrast agent application. Typical imaging parameters were as follows: Acquired voxel size = 2.0 × 2.0 × 8 mm^3^, reconstructed voxel size = 1.1 × 1.1 × 8 mm^3^, flip angle = 15° and effective echo time 4.75 ms.

Image analysis was performed offline using commercially available software (Medis Suite version 3.1, Medis Medical Imaging Systems bv Leiden, The Netherlands, and Extended MR WorkSpace version 2.6.3.5, Philips Medical Systems Nederland B.V., Best, The Netherlands). Late gadolinium enhancement (LGE) was assessed visually from mDIXON images. Segments with LGE were excluded from analysis, resulting in a total of 659 analyzed segments using a 16-segment-model. Mapping parameters were measured using QMap RE version 2.0 (Medis Medical Imaging Systems bv, Leiden, the Netherlands). Pre and post-contrast MOLLI images were manually corrected for in-plane-motion. The T1 and T2 relaxation times were calculated using nonlinear fitting with a maximum likelihood estimator (MLE). Extracellular volume (ECV) was calculated from pre and post-contrast T1-maps and the hematocrit, as described previously [6]. Due to possible harm, healthy controls received no contrast agent and were, therefore, not included in the ECV analysis. For comparison, data from a previous study using the same scanner model and contrast agent were used [14]. Exemplary images of patients with HFpEF, HFmrEF and HFpEF, and controls, are given in Figure 1A–L. For T1 native, T2 and ECV, the median value of all segments without late enhancement was calculated for each patient and used for all further analyses. In case of extensive artifacts in an imaging sequence, the patient was excluded from the respective analysis at the discretion of the analyzing physician. Peak left ventricular endocardial global longitudinal (GLS) and circumferential (GCS) strain were analyzed in accordance with a recent consensus document for the quantification of LV function using CMR [15]. Strain analysis included 2-chamber, 3-chamber and 4-chamber cine images and three preselected slices from the LV short-axis stack to correspond to basal, mid-ventricular and apical levels. The endocardial contours were drawn on the long and short-axis cine images with QMass version 8.1 (Medis Medical Imaging Systems bv, Leiden, the Netherlands) and were subsequently transferred to QStrain RE version 2.0 (Medis Medical Imaging Systems bv, Leiden, the Netherlands), where endocardial and epicardial borders were detected throughout the whole cardiac cycle using a tissue tracking algorithm. From these, global longitudinal and circumferential endocardial strain curves were calculated and the maximal amplitude was considered as the respective peak global strain. The strain ratio (SR = GLS/GCS) was calculated to assess for possible differences between heart failure groups.

Statistical analysis was performed using SPSS 24 (IBM, Armonk, NY, USA) and R 3.5.1 (The R Foundation for Statistical Computing, Vienna, Austria) [7]. Baseline data were reported as means ± standard deviations (SD) for interval and ratio-scaled parameters and as numbers and percentages for nominal and ordinal-scaled parameters. For comparisons between groups, ANOVA with Tukey–Kramer post-hoc analysis was performed. Pearson correlation coefficients were calculated for correlations between continuous variables and were tested for significance under the null hypothesis of *r* = 0. For quality of life, Spearman correlation was used. *p*-values below 0.05 were considered statistically significant.

Sample size calculations were performed for the detection of significant differences in native T1 relaxation time, T2 relaxation time and ECV between groups with a power of 80%. For native T1 relaxation time, previous studies have shown standard deviations between 20 and 50 ms and effect sizes of 30 to 40 ms [14,16]. For T2 relaxation time, standard deviations ranged from 3 to 7 ms and effect sizes from 3 to 5 ms [17,18]. For ECV, previous studies have shown standard deviations of 3–4% with effect sizes of about 4% [19,20]. Assuming a ratio of effect size to standard deviation of 1 for all three parameters, the minimum sample size is 16 per group.

## 3. Results

### 3.1. Patients

17 patients with HFpEF, 18 with HFmrEF, 17 with HFrEF and 17 controls were included in the analysis. The baseline data of the patients and controls are given in Table 1. Overall, controls were healthier, younger and more likely to be female than HF patients. Between HF groups, there were relevant differences in sex, age, coronary artery disease, history of smoking, lab values (most notably hematocrit and N-terminal pro brain natriuritic peptide—NT-proBNP) and medication use. NT-proBNP was log-normally distributed and transformed to logarithmic for correlation analysis. A correlation matrix of all continuous baseline and imaging parameters is given in Appendix A.

### 3.2. CMR-Parameters

#### 3.2.1. T2 Relaxation Time

One patient with HFpEF was excluded from analysis due to extensive artifacts. A boxplot of the measurements by group is given in Figure 2A. T2 relaxation times in HFmrEF and HFrEF patients were significantly elevated compared to healthy controls (Table 2). HFpEF patients did not differ significantly from healthy controls. Correlations of T2 with other MRI and baseline-parameters are given in Table 3. Scatter plots, and where appropriate, linear model parameter estimates are given in Figure 3 for the relationship between T2 and NT-proBNP, glomerular filtration rate (GFR), the 6 min walking test and quality of life.

#### 3.2.2. T1 Relaxation Time

One patient with HFpEF and one patient with HFmrEF were excluded from analysis due to extensive artifacts. A boxplot of the measurements by group is given in Figure 2B. Patients with both HFmrEF and HFrEF showed significantly higher T1 relaxation times than controls and HFpEF patients (Table 2). The difference between HFrEF and HFmrEF patients and the difference between HFpEF patients and controls was not statistically significant. The correlations of the T1 relaxation time with MRI and baseline-parameters are given in Table 3.

#### 3.2.3. Extracellular Volume (ECV)

The ECV was calculated for patients with HFpEF, HFmrEF and HFrEF. One patient with HFpEF and one with HFmrEF were excluded from analysis due to extensive artifacts in the T1 mapping sequences, from which ECV would be calculated. A boxplot of the measurements by group is given in Figure 2C and compared to historical data by Dabir et al. for volunteers [14]. None of the differences reached statistical significance (Table 2). The correlations of the ECV with MRI and baseline-parameters are given in Table 3.

#### 3.2.4. Strain

Boxplots for the measurements of GCS, GLS and SR (GLS/GCS) are given in Figure 2D–F. There were no statistically significant differences between HFpEF and controls. Patients with HFmrEF showed significant impairment in both circumferential and longitudinal strain, with further impairment being present in HFrEF patients, reflecting systolic dysfunction. The SR in HFpEF was significantly lower than in HFmrEF and HFrEF. The correlations of the strain parameters with MRI and baseline-parameters are given in Table 3.

## 4. Discussion

Our study is the first to date to examine advanced imaging markers of remodeling in patients with HFmrEF.

Differences in ECV did not reach statistical significance. The ECV values in our HFpEF patients were lower than those reported in other studies, ranging from 28.3% to 32.9% (Appendix A) [19,20,21,22,23]. The difference might be partly explained by differences in the applied scanners, sequences, contrast agents, image analysis, exclusion of LGE and LF-EF cut-off values for HFpEF. The lower ECV might also reflect our slightly healthier HFpEF group compared to previous studies. In our study, great effort was taken to manually adjust MOLLI-images for in-plane motion, thereby reducing artifacts by blood-signal. Blood has higher T1 and ECV values compared to myocardium, leading to falsely elevated myocardial ECV and T1 measurements in case of uncorrected in-plane motion. Additionally, while many studies used mid-ventricular septal measurements, our study measured the median value of all 16 myocardial segments, excluding segments with late enhancement. We chose our approach to be more representative of global left ventricular remodeling and less susceptible to artifacts and focal changes. One study comparing ECV in HFpEF versus HFrEF found a higher ECV in HFrEF, most likely representing advanced fibrosis in HFrEF [22]. Our study did not assess ECV in healthy volunteers. However, data from another study using the same scanner, mapping-sequence and contrast agent found a mean ECV of 27% ± 4% in healthy volunteers [14].

Native T1 relaxation times in HFmrEF were closer to HFrEF than HFpEF. A summary of studies examining native T1 relaxation times is given in Appendix A. One study using the same field strength and MRI manufacturer found higher T1 relaxation times for both controls and HFpEF patients compared to our data [19]. Another study using the same field strength and a different MRI manufacturer found similar T1 relaxation times [21]. Yet another study of healthy volunteers found lower T1 relaxation times compared to our controls [14]. The reason for the difference might be attributable to the local setup, as T1 relaxation times are known to be highly setup dependent [24]. Assuming that elevations in native T1 relaxation times truly reflect fibrosis, our findings suggest that HFmrEF shares common pathophysiological changes with HFrEF, while HFpEF seems to resemble a different pathophysiological entity. The lower values for fibrosis markers in HFpEF compared to HFrEF might at least partly explain the lower effectiveness of renin-angiotensin-aldosterone system (RAAS) inhibitors in HFpEF [3,25]. On the other hand, if our findings were to be confirmed in further studies, patients with HFmrEF would be expected to benefit from RAAS inhibitors. This would be in line with the findings of the PARAGON-HF trial, which showed no overall benefit of angiotensin-neprilysin inhibition in all patients with an LV-EF > 45% but a possible benefit for the subgroup with an LV-EF below the median [26].

T2 relaxation times were significantly elevated in HFmrEF and HFrEF compared to healthy controls. Our values for T2 relaxation times in controls were within the published reference range for our mapping sequence [27]. To our knowledge, no studies examining T2 relaxation time in HFpEF and HFmrEF have been published to date. An increase in T2 relaxation time is likely to reflect increased myocardial water content, as it is commonly seen in inflammatory states and myocardial edema [9,28]. Elevated T2 relaxation times have also been found in dilated cardiomyopathy and acute myocardial infarction [29,30]. Current evidence suggests a coincidence of myocardial inflammation and heart failure, although the exact mechanism remains unclear [31]. Consequently, our observed increase in myocardial T2 relaxation times might reflect subclinical myocardial inflammation in heart failure. Of all MRI parameters, T2 relaxation time showed the highest correlation with NT-proBNP and quality of life, and was the only MRI parameter significantly correlated with the 6 min walking test and the GFR. The latter warrants further investigation and might reflect cardiac involvement in renal disease. Irrespective of the exact underlying mechanism, our findings provide further evidence for pathophysiological similarities between HFmrEF and HFrEF.

The observed increase in both longitudinal and circumferential strain in HFmrEF and HFrEF reflects the reduced ejection fraction. Our strain values for controls and patients with HFpEF were within previously published reference values for cardiac MRI-derived endocardial strain [10]. Impairment of both GCS and GLS has been reported in patients with HFpEF both for MRI and speckle tracking echocardiography (STE) derived strain values. [32,33] Of note, while STE and MRI-derived measurements for GLS and GCS generally show good correlations, the absolute values may differ depending on the technique applied [34].

In our study, GLS and GCS seem to increase at different rates as the LV-EF decreases, so that the ratio of longitudinal to circumferential strain (strain ratio, SR) differs in HFpEF compared to HFmrEF and HFrEF. To our knowledge, no previous study has examined the strain ratio as a cardiac parameter. The significance of this remains to be determined in future studies.

One limitation of all parametric mapping studies is the inherent dependence of the measurements on the local setup. Our study does not intend to provide diagnostic criteria but to characterize the heart failure subtypes in relation to each other in terms of pathophysiological targets.

Another obvious limitation of our study is its small sample size. Consequently, small differences between groups and correlations between parameters may have been missed. Yet, despite the small sample size, we found significant differences between heart failure subgroups and hope to contribute towards the characterization of HFmrEF. Our *p*-values were not corrected for multiple testing and should be confirmed in further studies.

The groups differed in baseline parameters, such as age, sex, comorbidities, laboratory values and medication use. The differences observed are mostly consistent with previous studies, showing intermediate values in HFmrEF compared to HFpEF and HFrEF for the parameters age, sex and NT-proBNP [35,36,37]. Our HFpEF group seems to be slightly healthier compared to previous studies in regard to mean NT-proBNP, GFR and diuretic use, which might explain some of the differences. Regarding the prevalence of coronary artery disease, previous studies have shown mixed results, with some showing the highest prevalence in HFmrEF, and others in HFrEF [36,38,39]. Mapping parameters were not influenced by the presence of transmural LGE in the excluded segments (Appendix A). While mapping and strain parameters showed no relevant correlation with sex (Appendix A), hematocrit and age (Table 3), they were indeed highly correlated with NT-proBNP (Table 3). Due to the small sample size, no multivariate analysis was performed and our results may be confounded by these differences, which might be inherent to the underlying pathology.

The high number of patients reclassified with MRI-derived LV-EF compared to echocardiography-derived LV-EF diminishes the comparability of our results with trials relying on echocardiography alone. Furthermore, the question arises as to how many HFpEF patients in the general population actually do have a moderate reduction in LV-EF that is missed by echocardiography.

## 5. Conclusions

Despite the small sample size, our study was able to show significant adverse remodeling beyond systolic functional impairment in patients with HFmrEF that resembles the changes seen in HFrEF.

## Figures and Tables

**Figure 1 jcm-08-01877-f001:**
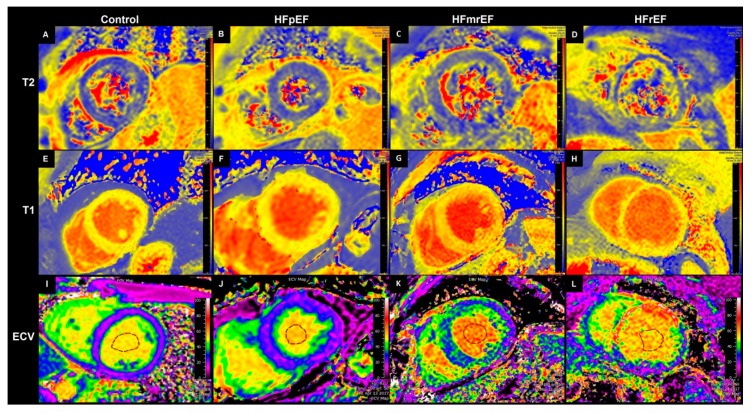
Exemplary medial short axis images of T2 relaxation time maps (first row, (**A**–**D**)), T1 relaxation time maps (second row, **E**–**H**) and extracellular volume (ECV) maps (third row, (**I**–**L**)). First column (**A**,**E**,**I**): Healthy control (ECV image from a patient from clinical routine, as no contrast agent was given to healthy controls in our study). Second column (**B**,**F**,**J**): Patient number 3 (HFpEF). Third column (**C**,**G**,**K**): Patient number 9 (HFmrEF). Fourth column (**D**,**H**,**L**): Patient number 11 (HFrEF). Segments with scars excluded from analysis. ECV = Extracellular volume. HFmrEF = Heart failure with moderately reduced Ejection fraction, HFpEF = Heart failure with preserved ejection fraction, HFrEF = Heart failure with reduced ejection fraction.

**Figure 2 jcm-08-01877-f002:**
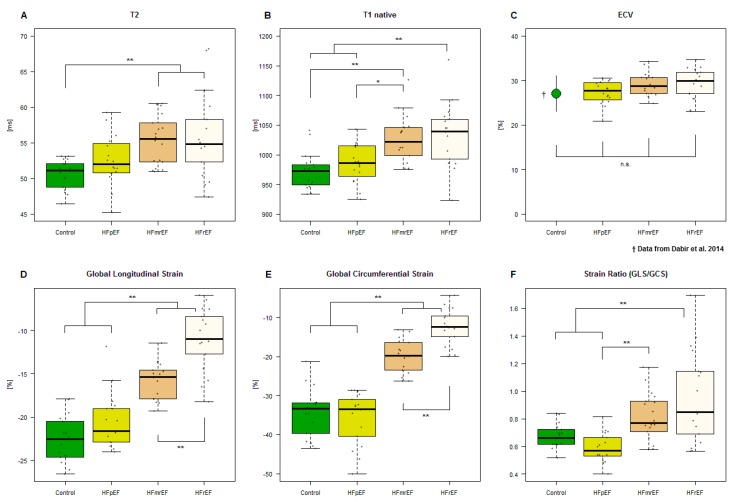
Boxplots of heart failure groups and controls versus (**A**) T2 relaxation time, (**B**) Native T1 relaxation time, (**C**) ECV (controls from Dabir et al. 2014) [14], (**D**) GLS, (**E**) GCS and (**F**) strain ratio (GLS/GCS). * Significant at α = 0.05. ** Significant at α = 0.01. n.s. = not significant at 0.05. GCS = global circumferential strain, GLS = global longitudinal strain. Other abbreviations as in Figure 1.

**Figure 3 jcm-08-01877-f003:**
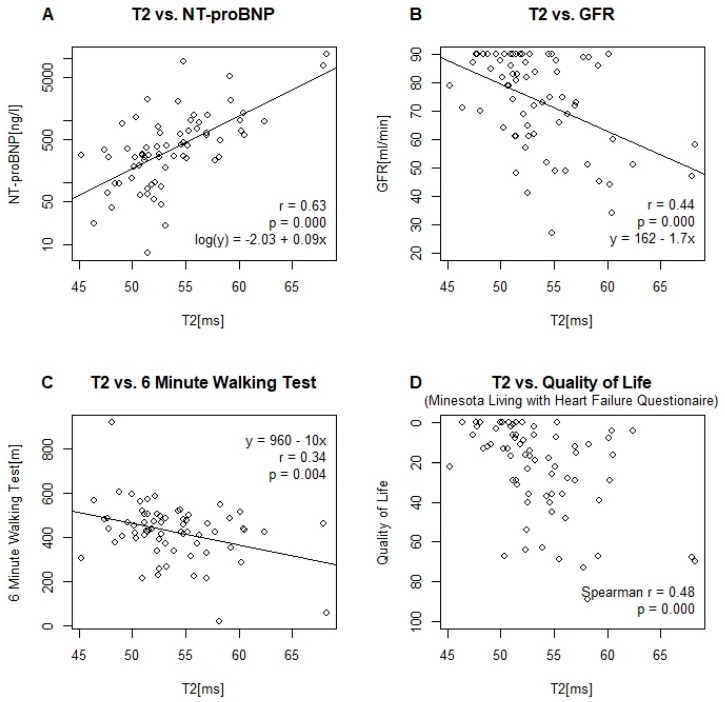
Scatter Plots and linear model parameters of T2 relaxation time versus (**A**) NT-proBNP (logarithmic scale), (**B**) glomerular filtration rate (GFR) and (**C**) 6 min walking test. (**D**) Scatter plot and Spearman’s correlation coefficient of T2 relaxation time versus quality of life, as assessed by the Minnesota Living with Heart Failure Questionnaire.

**Table 1 jcm-08-01877-t001:** Baseline characteristics.

Clinical Data		Control	HFpEF	HFmrEF	HFrEF
Female		8/17 (47%)	9/17 (53%)	6/18 (33%)	3/17 (18%)
Age	Mean ± SD	61.7 ± 8.5	78.1 ± 8.2	67.8 ± 9.0	64.4 ± 10.3
LVEF	Mean ± SD	63.8 ± 5.4	61.7 ± 6.1	44.7 ± 2.9	33.1 ± 4.8
LA (cm^2^)	Mean ± SD	19.5 ± 6.5	23.4 ± 4.9	22.8 ± 8.3	25.2 ± 6.6
RVEDD (mm)	Mean ± SD	31.5 ± 4.7	30.6 ± 3.8	29.6 ± 3.7	31.4 ± 5.4
Any LGE			7/17 (41%)	16/18 (89%)	15/17 (88%)
Transmural LGE			4/17 (24%)	7/18 (39%)	8/17 (47%)
Coronary Artery Disease		0/17 (0%)	11/17 (65%)	16/18 (89%)	11/17 (65%)
6 min Walking Test (m)	Mean ± SD	524 ± 126	345 ± 122	414 ± 88	414 ± 125
NYHA Class	2		9/17 (53%)	15/18 (83%)	12/17 (71%)
	3		8/17 (47%)	3/18 (17%)	5/16 (29%)
Quality of Life ^1^	Mean ± SD	5.0 ± 5.9	27.3 ± 22.8	28.3 ± 22.8	28.5 ± 24.9
Borg Score	Mean ± SD	7.5 ± 1.7	12.4 ± 2.4	10.67 ± 2.3	10.9 ± 2.5
Laboratory Values					
Hemoglobin (g/dL)	Mean ± SD	13.9 ± 1.1	12.8 ± 1.2	13.6 ± 1.1	15.0 ± 1.1
Hematocrit	Mean ± SD	0.40 ± 0.03	0.38 ± 0.03	0.40 ± 0.03	0.43 ± 0.04
Creatinin (mg/dL)	Mean ± SD	0.87 ± 0.20	0.92 ± 0.18	1.07 ± 0.33	1.09 ± 0.38
GFR (mL/min)	Mean ± SD	81 ± 10	71 ± 16	70 ± 18	72 ± 21
NT-proBNP (ng/L)	Mean ± SD	91 ± 62	614 ± 607	829 ± 1158	2257 ± 3447
Troponin T (ng/L)	Mean ± SD	7 ± 3	16 ± 12	19 ± 20	18 ± 12
CRP (mg/dL)	Mean ± SD	1.3 ± 1.4	2.9 ± 2.7	3.0 ± 4.2	1.0 ± 0.7
WBC (/nL)	Mean ± SD	6.1 ± 1.6	7.2 ± 2.4	8.5 ± 2.4	8.4 ± 2.3
Medication					
ACE-Inhibitors		2/17 (12%)	4/17 (24%)	7/18 (39%)	9/17 (53%)
Angiotensin-Receptor-Blocker		4/17 (24%)	11/17 (65%)	7/18 (39%)	8/17 (47%)
Calcium-Antagonist		4/17 (24%)	3/17 (18%)	3/18 (17%)	1/17 (6%)
Mineralocorticoid-Receptor-Antagonist		0/17 (0%)	2/17 (12%)	4/18 (22%)	11/17 (65%)
Angiotensin-Receptor-Neprilysin-Inhibitor		0/17 (0%)	0/17 (0%)	0/18 (0%)	4/17 (24%)
Beta-Blocker		6/17 (35%)	10/17 (59%)	14/18 (78%)	17/17 (100%)
Statin		2/17 (12%)	8/17 (47%)	15/18 (83%)	11/17 (65%)
Thiazide Diuretic		4/17 (24%)	4/17 (24%)	2/18 (11%)	1/17 (6%)
Loop Diuretic		0/17 (0%)	3/17 (18%)	7/18 (39%)	6/17 (35%)

Discrete values given as absolute number and percentage of respective HF group. Continuous values given as mean and standard deviation. ACE = angiotensin-converting-enzyme. CRP = C reactive protein. GFR = glomerular filtration rate. LA = left atrium. LGE = late gadolinium enhancement. LVEF = left ventricular ejection fraction. NT-proBNP = N-terminal pro brain natriuritic peptide. NYHA = New York Heart Association. RVEDD = right ventricular end diastolic diameter (MRI, three chamber view). WBC = white blood cell count. ^1^ Minnesota Living with Heart Failure Questionnaire.

**Table 2 jcm-08-01877-t002:** Group means and *p*-values for group-wise comparisons.

	Heart Failure Group				*p*-Value				
	Control	HFpEF	HFmrEF	HFrEF	Controls vs. HFpEF	Controls vs. HFmrEF	HFpEF vs. HFmrEF	HFpEF vs. HFrEF	HFmrEF vs. HFrEF
T2 (ms)	50.6 ± 2.1	52.6 ± 3.6	55.4 ± 3.4	56.0 ± 6.0	0.499	0.005 **	0.190	0.078	0.967
T1 native (ms)	972 ± 31	985 ± 32	1027 ± 40	1033 ± 54	0.776	0.001 **	0.023 *	0.005 **	0.954
ECV (%)	27 ± 4 ^†^	27.3 ± 2.6	29.2 ± 2.6	29.3 ± 3.4	0.993	0.186	0.303	0.271	>0.999
GLS (%)	−23.0 ± 3.5	−20.8 ± 3.9	−15.7 ± 2.1	−11.0 ± 3.6	0.252	<0.001 **	<0.001 **	<0.001 **	<0.001 **
GCS (%)	−34.5 ± 6.2	−35.8 ± 6.7	−19.9 ± 4.1	−12.4 ± 4.6	0.902	<0.001 **	<0.001 **	<0.001 **	0.001 **
SR	0.68 ± 0.09	0.59 ± 0.11	0.82 ± 0.17	0.96 ± 0.33	0.600	0.159	0.007 **	<0.001 **	0.151

Data reported as means ± standard deviations. *p*-values calculated from one-way ANOVA with Tukey–Kramer post-hoc analysis. GCS = global circumferential strain, GLS = global longitudinal strain, ECV = extracellular volume, SR = strain ratio (GLS/GCS), T1 = T1 relaxation time and T2 = T2 relaxation time; other abbreviations as in Table 1. * Significant at α = 0.05. ** Significant at α = 0.01. ^†^ Data from Dabir et al. 2014

**Table 3 jcm-08-01877-t003:** Correlations of parameters.

		T2	ECV	T1 Native	GLS	GCS	SR	**LV-EF**
ECV	Pearson r	0.353 **						
	*p* Value	0.010						
T1 native	Pearson r	0.660 **	0.472 **					
	*p* Value	<0.001	<0.001					
GLS	Pearson r	0.351 **	0.294 *	0.518 **				
	*p* Value	0.003	0.034	<0.001				
GCS	Pearson r	0.372 **	0.256	0.484 **	0.868 **			
	*p* Value	0.002	0.067	<0.001	<0.001			
SR	Pearson r	0.309 **	0.062	0.308 **	0.353 **	0.698 **		
	*p* Value	0.009	0.663	0.010	0.003	<0.001		
LV-EF	Pearson r	−0.422 **	−0.242	−0.518 **	−0.882 **	−0.929 **	−0.614 **	
	*p* Value	<0.001	0.090	<0.001	<0.001	<0.001	<0.001	
Age	Pearson r	0.234	−0.100	0.064	−0.199	−0.270 *	−0.202	0.246 *
	*p* Value	0.052	0.483	0.603	0.095	0.023	0.091	0.020
log(NT-proBNP)	Pearson r	0.642 **	0.287 *	0.601 **	0.544 **	0.544 **	0.234 *	−0.538 **
	*p* Value	<0.001	0.039	<0.001	<0.001	<0.001	0.050	<0.001
Glomerular filtration rate	Pearson r	−0.441 **	0.106	−0.123	−0.065	−0.057	−0.054	0.085
	*p* Value	<0.001	0.455	0.314	0.589	0.637	0.655	0.487
C-reactive protein (mg/dL)	Pearson r	0.105	−0.154	−0.006	−0.044	−0.087	−0.112	0.062
	*p* Value	0.393	0.281	0.961	0.715	0.473	0.357	0.614
Hematocrit	Pearson r	0.101	−0.180	0.150	0.330 **	0.405 **	0.343 **	−0.471 **
	*p* Value	0.407	0.201	0.219	0.005	<0.001	0.003	<0.001
6 min walking test	Pearson r	−0.345 **	0.273	-0.150	−0.102	0.014	0.138	0.037
	*p* Value	0.004	0.055	0.224	0.403	0.912	0.258	0.765
Quality of life	Spearman r	0.484 **	0.187	0.359 **	0.184	0.168	0.097	−0.230
	*p* Value	<0.001	0.185	0.002	0.124	0.162	0.420	0.058

*p*-values are given for the significance of the correlation coefficient. NT-proBNP was transformed to logarithmic scale. Abbreviations as in Table 1 and Table 2. * Significant at α = 0.05. ** Significant at α = 0.01.

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
