# Peer review of "CMR Tissue Characterization in Patients with HFmrEF"

_jcm, 2019, doi:10.3390/jcm8111877_

Round 1

Reviewer 1 Report

Doeblin and coworkers provide a CMR based characterization of  HFpEF, HmrEF, and HFrEF.

I congratulate the authors for studying HFmrEF using state of the art CMR techniques including parametric mapping and strain analysis. Several aspects, however, merit comment.

My biggest concern is the limited study size. The study was designed to include 90 patients in total, so I have concerns to present preliminary results in fewer patients to perform a phenotypic characterization. Especially parameters such as ECV are unlikely to show significant differences in such a small cohort. MRA usage of 12% in HFpEF is surprisingly low to me. More so, loop diuretic treatment in 18% of HFpEF although 47% were in NYHA III is inconsistent with daily clinical practice experience. Please comment. HFpEF were >10 years older as HFrEF and controls which makes comparison between CMR parametric parameters difficult if not correcting for age. And a multivariable correction would not make sense in this sample size. As a member of the group co-authored the recently published PARAGON trial, I suggest to include and briefly discuss its findings as HFmrEF patients were also included in PARAGON. Reference 11 in the text does not fit Reference 11 in the Reference section. As also depicted in Figure 1, the majority of HFrEF patients seems to be of ischemic nature. This makes a comparison between other entities difficult since also in non-infarcted areas parametric alterations are known to occur. How do results differ when only looking at HFrEF without extensive post-MI scar? Especially for T1 and even more so for T2 times. GLS values in HFpEF were not different from controls. However, one study showed markedly decreased GLS values in a different HFpEF cohort (10.1016/j.jcmg.2019.02.016) although they did not compare with controls. I assume HFpEF patients in the current manuscript rank among healthier subjects, as I also assume based on the preserved renal function (mean GFR >70) and low rate of diuretic treatment. This should be clearly stated in the limitation section. I am missing results for CMR parameters. Were there differences in right ventricular function and left atrial size? This is of crucial importance in all HF entities. How many patients had LGE, to what extent? How were patients recruited? Based on echo results? There it would be interesting to see, especially at this sample size, how many patients changed the respective HF group when comparing LVEF results from echo with CMR.

Reviewer 2 Report

This is an interesting study that addresses an important problem – that of the “orphan” class of HFmrEF. In this small study comparing controls, HFpEF, HFmrEF and HFrEF, the authors conclude that measures of global strain and T1 and T2 relaxation times were abnormal in HFmrEF and HFrEF while ECV volume was not.

As the authors indicate, it was surprising that ECV was not different in any of the HF groups compared to controls. In addition to the issue of not being able to replicate increased ECV in HFpEF and HFrEF,  ECV would have given more information about underlying biology compared to changes in global systolic function. While the findings of T1 relaxation time seem to indicate higher values in HFmrEF, it is unusual that HFrEF due to wide range of values does not show consistent difference form other groups.  Hence overall with these small numbers of participants, I am not sure that firm conclusions can be made abpout the tissue characteristics in HFmrEF.

It might be useful also to correlate MR parameters with blood markers of remodeling and fibrosis

The title may be more appropriate as a comparison of all types for HF than focus on HFmrEF.  

Round 2

Reviewer 1 Report

The authors addressed most issues. 

However, it is not acceptable to reply "our analysis was focused on left ventricular parameters, our research group is analyzing other parameters and will report their findings in the near future". Data on atrial size and right ventricular size and function. These parameters are for sure already available and even if they would have to be measured: it would take less than a day for all measurements in this small cohort!

You do not have to focus on these parameters and I understand that you aim to publish separate projects with an emphasis on atrial size and function as well as RV size and function. Nonetheless, basic information must be included. 

Author Response

Dear Reviewer,

we have included LA and RV size as requested. Regarding RV-Function, we are planning on publishing a separate paper on RV remodeling and do not wish to include this information in the current paper. For your information, the values for RV-EF were as follows:

HF_Group RV_EF
Control Mean 58,1
SD 3,6
HFpEF Mean 57,3
SD 6,9
HFmrEF Mean 52,7
SD 11,0
HFrEF Mean 57,8
SD 10,3

Reviewer 2 Report

Authors have responded to the critiques with additional analyses

Author Response

We thank the Reviewer for his time and effort.